# Changes in Motor Competence of 4–8-Year-Old Children: A Longitudinal Study

**DOI:** 10.3390/ijerph21020190

**Published:** 2024-02-07

**Authors:** Pim Koolwijk, Ester de Jonge, Remo Mombarg, Teun Remmers, Dave Van Kann, Ingrid van Aart, Geert Savelsbergh, Sanne de Vries

**Affiliations:** 1Research Group Healthy Lifestyle in a Supporting Environment, The Hague University of Applied Sciences, 2501 EH The Hague, The Netherlands; e.a.l.dejonge@hhs.nl (E.d.J.); s.i.devries@hhs.nl (S.d.V.); 2Institute of Sport Studies, Hanze University of Applied Sciences, 9747 AS Groningen, The Netherlands; r.mombarg@pl.hanze.nl (R.M.); i.van.aart@pl.hanze.nl (I.v.A.); 3School of Sport Studies, Fontys University of Applied Sciences, 5644 HZ Eindhoven, The Netherlands; t.remmers@fontys.nl (T.R.); d.vankann@fontys.nl (D.V.K.); 4Department of Behavioural and Human Movement Sciences, Section Motor Learning & Performance, Vrije Universiteit (VU) Amsterdam, 1081 HV Amsterdam, The Netherlands; g.j.p.savelsbergh@vu.nl; 5Department of Public Health and Primary Care, Health Campus the Hague, Leiden University Medical Centre, 2511 DP The Hague, The Netherlands

**Keywords:** early childhood, longitudinal, middle childhood, motor quotient trajectories, undesirable patterns

## Abstract

Objectives: The development of children’s motor competence (MC) from early to middle childhood can follow different courses. The purpose of this longitudinal study was to describe and quantify the prevalence of patterns of MC development from early to middle childhood and to identify undesirable patterns. Design: The study used a longitudinal design. Data were collected in three consecutive years, between February 2020 (T0) and May 2022 (T2). Methods: A total of 1128 typically developing Dutch children (50.2% male) between 4 and 6 years old at baseline (M = 5.35 ± 0.69 years) participated in this study. MC was measured with the Athletic Skills Track and converted into Motor Quotient (MQ) scores. To convert all individual MQ scores into meaningful patterns of MC development, changes in MQ categories were analyzed between the different timepoints. Results: A total of 11 different developmental patterns were found. When grouping the different patterns, five undesirable patterns were found with 18.2% of the children, showing an undesirable pattern of MC development between T0 and T2. The patterns of motor development of the other children showed a normal or fluctuating course. Conclusions: There is a lot of variation in MC in early and middle childhood. A substantial percentage of young children showed undesirable MC developmental patterns emphasizing the need for early and targeted interventions.

## 1. Introduction

Childhood is a critical time for the development of motor competence (MC). MC enables children and adolescents to successfully participate in various types of physical activity (PA) [1]. This is a global term to describe goal-orientated human movement [2] and can be defined as an individual’s degree of proficiency in performing a wide range of motor skills as well as the mechanisms underlying this performance (e.g., motor control and coordination) [3]. Considering that MC at a young age contributes to the development of an active lifestyle [4,5], it is imperative to assess and monitor MC, particularly in early- and middle childhood. Clark and Metcalfe [1] proposed a well-recognized model to explain motor behaviors across a lifespan. This ‘mountain of motor development’ includes an ascent of six passages of motor development starting from birth (the reflexive period) and ending at the compensation period as a time when the system adapts or compensates for detrimental changes in organism constraints. Each period builds on the skills learned in the previous period. Motor development is related, but not strictly dependent on age. So, the time spent in each stage of development varies for each individual depending on factors that determine skill acquisition (e.g., the amount of instruction, the quality of instruction, individual qualities) [6]. Parallel to motor development, young children undergo significant biological development across various domains. During the transition from early to middle childhood, children experience steady growth, with an average annual increase in height and body mass. Due to improvement in coordination, fine motor skills (e.g., handling smaller objects, writing, and drawing) are performed with increasing proficiency. Also, neural connections are strengthened, supporting cognitive functions.

Without early experiences, children would be limited in their ability to explore context-specific movement solutions [7]. This limitation can be of danger for the health status of young children since MC has an inverse relationship with developing obesity [6,8,9]. Based on the proficiency barrier introduced by Seefeldt in 1980, not reaching a basic level of fundamental motor skills may restrict children from participating in sport activities, which is considered undesirable. It has been hypothesized that in early childhood (EC) (2–5 years), MC and PA levels display great variability and that during middle childhood (MCD) (6–9 years), this existing relationship is reinforced due to environmental constraints (e.g., parenting style, experiences, opportunities to play, demographics, and social factors) already present at an early age [10,11,12]. As a result, children with higher levels of MC in childhood have opportunities to participate in a broader and more complex repertoire of physical activities later in life, also leading to a reinforcement of mediating factors (e.g., enjoyment, perceived competence, etc.). In contrast, experiencing lower levels of MC leads to a negative spiral of disengagement in PA and its mediating factors. Studies have shown that these children may be at risk of developing overweight [11,13].

Although there are historical data regarding the development of health-related fitness variables and PA in EC, trends of MC development within this age group are very limited [14]. However, there seems to be a negative trend over the last five decades regarding group mean levels of MC at an early age [14,15]. It should be noted that this decline in MC differs between the type of fundamental movement skill (FMS) tested [16]. In recent literature, most evidence regarding children’s MC in early and middle childhood has been provided by systematic reviews [17,18,19,20] or by cross-sectional studies [21,22,23,24]. Only a few studies have used a longitudinal design with at least two [25,26,27] or more [28] follow-up measurements of MC in early childhood. Using longitudinal data is essential for understanding the dynamic and temporal direction of MC development, especially in young children. As indicated, only a few longitudinal studies have been conducted to describe the development of MC in EC and/or the transition to MCD. In a longitudinal study performed by Estevan [26], more than one hundred young children in the transition of EC/MCD, from 4 to 9 years old, were profiled to detect actual and perceived MC trajectories. Regarding the actual motor competence, object control skills tended to increase over time for boys but decreased over time for girls between 7 and 9 years old. An earlier study with respect to the transition phase of EC/MCD by Bardid and colleagues [29] also showed gender differences in favor of boys regarding object-control skills. Coppens and colleagues [30] reported that MC generally increased over time in children in middle childhood from 6 to 9 years old with individual variations. There was no significant variance between low, average, and highly motor-skill-developed children, and although the gender differences were unclear, in this study, girls made less progress in MC than boys. In contrast with previous studies, Rodrigues [31] tracked MC and physical fitness starting in middle childhood (age six) for four consecutive years. They noticed that not all children improved in MC across childhood years, and some even regressed over time. In the same study conducted by Rodrigues [31], the object-control skills of girls lagged behind those of boys as they grew older. In general, there seems to be a tendency of increasing MC as children grow older, while girls seem to lag behind in object-control-skill development.

Longitudinal data on MC patterns in children maturing from early to middle childhood are scarce. To our knowledge, individual patterns have only been reported in studies in middle childhood [30,31]. Start(V)aardig (Dutch for ‘Skillful Start’) was a four-year project containing three subsequent measurements of MC with time intervals of one year each. Taking into account that children’s MC can remain stable, increase, or decrease at each of these time intervals, theoretically, 11 different patterns of MC development could occur. Therefore, the objectives of this three-year longitudinal study were to quantify the prevalence of these different patterns of MC development among a sample of school-aged children in transition from early to middle childhood and to identify undesirable patterns. Although MC levels vary in EC, we expected an overall increase in MC since MC development is positively stimulated as a result of appropriate practice, encouragement, feedback, and instruction [10]. Also, since children generally have positive perceptions of their MC in EC, a window of opportunity is provided for stimulating MC development in the transition from EC to MCD [32,33]. Our hypothesis was that patterns of MC development reflecting an increase between EC and MCD would be the most prevalent. We expected that the prevalence of undesirable patterns would be relatively low. However, for policy makers and sport professionals, it is crucial to find out which part of the study population has an undesirable pattern of MC development.

## 2. Methods

### 2.1. Study Sample and Data Collection

A longitudinal design was used to study 1578 typically developing children (50.2% boys) aged between 4 and 6 years at baseline. The study population included children attending between grades one and two (between 4 and 6 years old) of the Dutch public school system. In total, 36 primary schools participated in the study across three regions of the Netherlands. The participating schools were internship schools of the connected universities representing urban and rural areas of the Netherlands. The population of the schools had a wide range of social economic positions. Written informed consent was given by parents/guardians of all participants (63% consent rate). Exclusion criteria included having less than three valid assessment points, being motor or physically impaired, and not complying with the written informed consent. Prior to the study onset, ethical approval was obtained by the Ethics Committee of the Faculty of Behavioral and Movement Sciences, VU University, Amsterdam, the Netherlands (ref. number VCWE-2019-139).

The study used a longitudinal prospective cohort design (see Figure 1). Data were collected at three subsequent timepoints with a one-year interval between February 2020, timepoint 0 (T0) and May 2022, timepoint 2 (T2). Due to temporary absences (e.g., illness) or permanent origin (e.g., moving to another school), 186 children (11.8%) of the initial study population at T0 dropped out one year later and 261 children (18.8%) dropped out after two years. Demographic information (e.g., child’s sex and date of birth) was parent-reported via a survey at study enrollment. To study MC patterns, we enrolled all children with three valid measurements of MC (complete MQ data). The present study population consisted of 1128 children (M = 5.35 ± 0.69 years).

### 2.2. Procedures

All participants wore light sport clothing and were barefoot during testing. Trained research assistants assessed the children during the regular physical education lessons. All research assistants participated in a two-hour training session to conduct the measurements according to the standardized testing guidelines.

#### 2.2.1. Measuring Motor Competence

MC was measured with the Athletic Skills Track (AST), a validated product-orientated assessment tool for primary-school-aged children [34]. The AST-1 was conducted for the youngest children in the age range of 4–6 years; the AST-2 was performed for the age group of 6–9 years old. The tracks included a string of different FMSs (AST-1: *n* = 5, AST-2: *n* = 7) to be completed as quickly as possible. The test–retest reliability of the AST was proven to be high (ICC = 0.881 (95%), CI: 0.780–0.934) for the AST-1 and 0.802 (95% CI: 0.717–0.858) for the AST-2 in a sample of 4–12-year-old Dutch children [35]. The internal consistency of the AST was above the acceptable level of Cronbach’s α > 0.70 (α = 0.764) [35,36], and there was a moderate to high correlation between the time to complete the AST-1 and AST-2 with the age- and sex-related motor quotients of the Körperkoordinations Test für Kinder (AST-1: r = −0.747, *p* = 0.01 and AST-2: r = −0.646, *p* = 0.01) [35].

#### 2.2.2. Other Measurements

Participants’ body height and mass were individually measured to the nearest 0.1 cm using a stadiometer (SECA 217, Hamburg, Germany) and to the closest 0.1 kg using a digital scale (SECA 878dr, Hamburg, Germany) at the start of the test session.

### 2.3. Data Analysis

MC was expressed in age- and gender-specific Motor Quotient (MQ) categories. MQ categories were derived from the time to complete the Athletic Skills Track (AST time) using age- and gender-specific norm values described by Hoeboer and colleagues [34]. In brief, an AST-time below the 25th percentile of AST norm values was classified as ‘low’, an AST time between the 25th and the 75th percentile of the norm values was classified as ‘normal’, and an AST time above the 75th percentile of the norm values was classified as ‘high’. Subsequently, patterns of MC development were defined based on the changes in MQ categories between T0 and T1 and between T1 and T2. Table 1 provides an overview of all classifications of motor development. ‘Undesirable’ patterns were classified based on a combination of the course and the MQ category. In brief, the MQ category may remain stable, continuously change in a certain direction (i.e., increase or decrease), or fluctuate over time.

### 2.4. Classification of Undesirable Patterns

Obviously, a stable but low MQ category is considered undesirable (pattern a), as is a continuous decrease over time (pattern d). Regarding the fluctuating patterns, we decided to classify some of these to be undesirable based on the assumption that a decrease between T1 and T2 was likely to be a more serious signal of undesirable development than a decrease between T0 and T1. Our motivation underlying this assumption was that a decrease between T0 and T1 could have been affected by the restrictions in response to the COVID-19 pandemic in the Netherlands, whereas a decrease between T1 and T2 could not have been. Additionally, we assumed that a sudden decrease at a slightly older age (i.e., MCD) was a more serious predictor of problems later in life than a decrease at a younger age [37]. Finally, we only considered a decrease to be undesirable when it led the MQ category to drop below the normal value at T2. Therefore, a decrease before or after stabilization (pattern f and h) or after an increase (pattern g) was considered ‘undesirable’ but only when the MQ categories at all timepoints did not exceed the normal category. Undesirable patterns were based on the prerequisite that every child must possess a basic motor skill level to participate in sports. A decline or failure to reach this level was considered to be undesirable. We will further refer to the selected patterns as ‘undesirable’ and to the children showing these patterns as ‘potential target groups’.

The anthropometric measures of body mass index (BMI) were derived by dividing the children’s body mass in kilograms by their height in meters squared. Data analysis was performed with the Statistical Package for the Social Sciences (SPSS version 27.0, 64-bits edition, SPSS Inc., Chicago, IL, USA). Visualizations were obtained with R studio (version 4.2.2, R Foundation for Statistical Computing, Vienna, Austria).

## 3. Results

Baseline characteristics are presented in Table 2. In this table, descriptives of the study population (i.e., calendar year, school grade, age, BMI, childhood phase, and sex) are presented for each timepoint (T0, T1, and T2). At baseline, the vast majority (81%) of children were in the EC phase, whereas at T2, almost all the children (97%) reached MCD.

### Prevalence of Different MC Development Patterns

Figure 2 shows the longitudinal MQ data of the study population. According to Hoeboer and colleagues [34], MQ scores ≤40 and ≥180 were detected as outliers (*n* = 3) and therefore removed before analysis. In total, 11 different patterns were predefined.

Of those 11 patterns, the ‘continuously normal’ pattern was the most prevalent pattern in our study population, followed by all variations of the fluctuating patterns. Both continuously decreasing and increasing patterns were rare.

In total, 18.2% of the children showed an undesirable pattern of MC development from T0 to T2 (patterns: a (4.5%); d (1.2%); f (2.2%); g (4.4%); and h (5.9%)), of which the pattern reflecting a decrease after stabilization (h) was the most prevalent. These undesirable patterns are indicated in Figure 2 with the red lines. Post hoc analyses showed that the boy–girl ratio was relatively equally distributed in most of the patterns, although within pattern b (continuous decrease), boys were underrepresented (29% boys) and within pattern k (increase after stabilization), boys were overrepresented (63% boys). An examination of dropouts indicated no notable differences in MQ distribution between those who completed all measurements and those who dropped out at T1 or T2, indicating that the children included in our analysis properly represented the initial study population of ‘Start(V)aardig’.

## 4. Discussion

The purpose of this longitudinal study was to describe and quantify the prevalence of 11 predefined patterns of MC development from early childhood to middle childhood. Of these 11 patterns, five were classified as ‘undesirable’. The majority of the children had a stable ‘normal’ or increasing ‘high’ development of MC. However, the undesirable patterns found represented 18.2% of the study population. As stated by Stodden [11] later revised by lower levels of MC at a young age can lead to a negative spiral of disengagement in PA and its mediating factors and increases the risk of developing overweight.

It has been hypothesized that young children have their own calendar regarding MC development [38]. This individuality refers not to sequence of motor skill learning but more to the rate and extent of motor skill acquisition. Due to environmental constraints (e.g., parenting style, experiences, opportunities to play, demographics, and social factors), great variation in MC in EC is expected [11,13]. This variation in MC was seen in our study with children scoring low, normal, or high at T0. Our study also proved that, as children grow older, MC development varies widely between children. This was in line with a longitudinal study conducted by Coppens [30] who also found variations in MC development of 6–9-year-old children. However, our findings were in contrast with two other studies [29,39]. Bardid [29] followed a Flemish population of 3–8-year-old children, which showed increased proficiency in MC as children grew older. Similar results were found by Reyes [39] with a notable rate of change in MC development from the age of six, in favor of boys performing better than girls. Studies on MC development related to PA and fitness during early childhood [27,40] or during middle childhood [26,41] both emphasized that MC is an important precursor to PA and fitness and is thus important for sport participation as children grow older. It is therefore expected that children would increase in MC during middle childhood. However, our longitudinal study results showed that a considerable percentage of children developed an undesirable pattern of MC, even as they grew older. A possible explanation for this negative course could be an increase in sedentary behavior due to social media even at a young age. Study results of measurement performed in 2014 showed that children from urban, low-income, and minority communities had almost universal exposure to mobile devices, and most had their own device by age four [42]. Excessive early childhood screen media use is associated with poorer MC, especially among children with long-term exposure [43]. Also, the daily living environment of young children should be considered when explaining negative courses of MC development. A Finnish study revealed that residential density was related to children’s MC and their outdoor play behavior, even during EC [44].

Undesirable patterns could also be the result of long-term governmental policies regarding physical education (PE) at schools. In the Netherlands, PE lessons (90 min per week) during EC (grade 1 and 2) are mainly given by group teachers, whereas PE during MCD (grade 3 and 4) is given by PE professionals. Some sport clubs offer memberships for 5-year-old children; however, in general, structured training is provided from 6 to 7 years old. Refrainment from PE and sports during the COVID-19 pandemic could have prevented motor development. All participants in the current study experienced a three-month lockdown from 16 March to 11 May 2020, and during January 2021 with restrictions on PA at sport clubs remaining in effect from 16 March to July 2020 [45]. Within our results, three patterns (d, f, and j) showed a decrease in MQ scores between T0 and T1 (at the time of lockdown restrictions). To what extent the restrictions had an influence on the children showing this negative course remains unclear. The authors consider it unlikely that the development of children with a ‘continuous decrease’ (pattern d) were affected by lockdown effects. However, it should be mentioned that the prevalence reported in this study might have slightly overestimated the ‘true’ prevalence adjusted for potential effects of the lockdown. Results from a Japanese study performed on EC children [46] suggested that the COVID-19 pandemic impeded the development of FMSs, especially those for object-control skills. Studies performed on MCD-aged children showed that the COVID-19 pandemic resulted in decreased motor skills due to children’s activities at home, such as increased sleeping habits, increased eating habits, and increased screen time [47,48,49]. However, a recently published study by den Uil and colleagues [50] showed the COVID-19 lockdowns in the Netherlands did not negatively affect motor skill development of MCD children. Therefore, more longitudinal research is needed to explore the long-term effects of the COVD-19 pandemic on MC development. The present study is, to our knowledge, one of the few longitudinal studies analyzing MC development starting from early childhood on such a large scale. The distribution of the MQ categories measured in our study sample between 2020 and 2022 were representative of the national MC screening in which the same assessment tool was used [51]. Besides the number of participants included, the detailed representation of the different patterns gave a clear insight into MC development during the transition from EC to MCD. Furthermore, the patterns defined in this study were based on a validated categorization of MQ scores by Hoeboer [34].

Even though this study had certain strengths, it also had limitations. Methodological difficulties occurred when measuring children in the EC phase (e.g., type of assessments used, concentration of children when being tested, etc.) and might have influenced our results. However, well-trained research assistants and a validated assessment tool were valuable in addressing these issues. One of the goals of this study was to present a clear course of MC development during the transition from EC to MCD. As shown in Table 2, our study population at T1 was not ideally distributed (33% EC and 67% MCD), so caution is needed in interpreting the results during the transition phase. Despite the dropouts, children who left the study did not differ from the participants included in their MC category. This would suggest that the dropouts did not bias the study results. Although the number of children included in this study was high, after converting the MQ scores into meaningful patterns, the group sizes were not big enough to test for a relationship between sex and EC/MCD phase. Therefore, only the boy/girl ratio was presented.

Our assessment tool, the Athletic Skills Track, was a product-orientated assessment tool focused on mastering two of the three FMSs (i.e., locomotor and balance skills). The third FMS (i.e., object-control skills) was not part of the test protocol and was not evaluated [51]. Since locomotor skills normally develop earlier compared with object-control skills [52], the undesirable patterns found in this study were even more worrying. Individuals during EC without mastering FMSs may require more practice than a child with experience in order to achieve a proficient skill level [7]. In addition, without early, developmentally appropriate experiences, children will be limited in exploring context-specific movement solutions that enhance mobility in more complex movement settings (e.g., you have to be able to run before taking part in soccer, tennis, or basketball). School PE lessons and/or other activities in the domain of sports are needed to improve children’s MC and increase PA for optimal health. Identifying undesirable patterns of MC at a young age would enable sport and healthcare professionals to adequately intervene and create prospectives for lifetime sport participation [53]. Since 2010, the number of FMS interventions has been rapidly increasing, indicating the need for stimulating MC development during EC. However, it should be evaluated if youth sports programs afford appropriate interventions to remedy the proficiency barrier for potential target groups or if youth sports programs increase the barrier and contribute to the discontinuation of sport participation in sport and physical activity [54]. A recently conducted systematic review to evaluate the effectiveness of FMS intervention during EC showed that more retention results are needed to evaluate the long-term effect of these interventions [55].

Besides developing targeted high-quality interventions, more longitudinal studies are needed to determine which modifiable determinants (e.g., BMI, perceived MC, enjoyment, etc.) influence MC development in young children [10]. It is also recommended to consider other contextual factors to better understand and support MC development during the transition from EC to MCD. One such factor is socioeconomic position, which may be positively associated with MC development [18,25]. Finally, longitudinal studies conducted in understudied countries taking into account different environmental and cultural contexts are warranted to gain more insight into global trends of MC development [10].

## 5. Conclusions

This study proved that MC does not automatically increase as children grow older. The prevalence of 11 predefined developmental MC patterns showed that most of the children displayed a stable ‘normal’ or fluctuating pattern with ‘normal to high’ development of MC. However, five undesirable patterns were identified representing almost twenty percent of the participants. Children with an undesirable pattern of MC development could be labelled as potential target groups for motor intervention. The outcome of this study emphasized the need for early and targeted motor interventions. Also, it is recommended to identify modifiable determinants that are associated with undesirable patterns of MC in order to detect children at risk as early as possible and help them improve their motor skills.

### Practical Implications

Physical education (PE) teachers should be aware that motor competence development widely varies during the transition from early to middle childhood.More emphasis should be placed on implementing targeted motor interventions focused on potential target groups to avoid the risk of developing an unhealthy, inactive lifestyle.Modifiable determinants associated with undesirable patterns of motor competence should be better understood.

## Figures and Tables

**Figure 1 ijerph-21-00190-f001:**
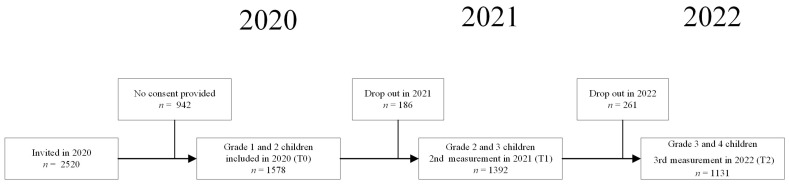
Flow diagram of study population.

**Figure 2 ijerph-21-00190-f002:**
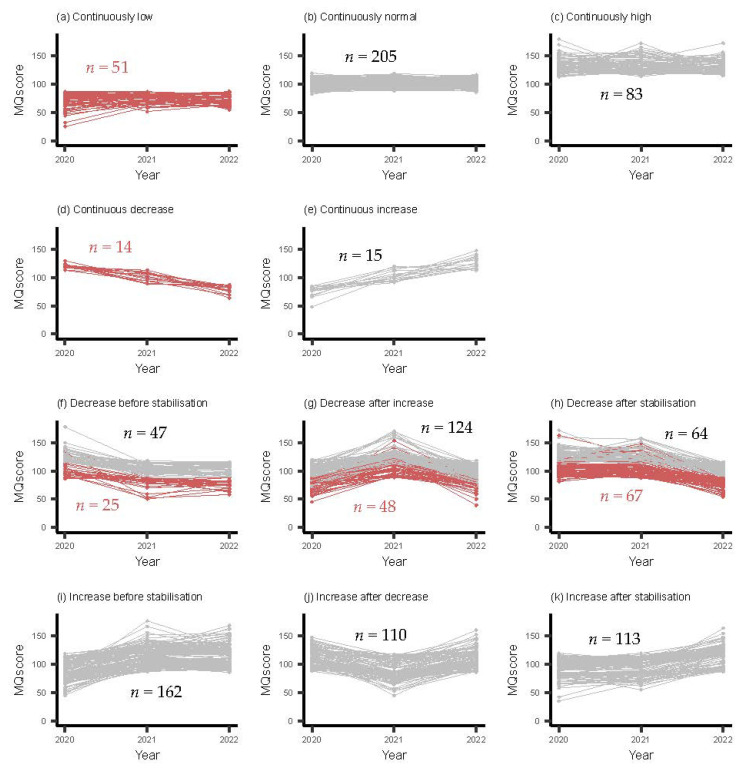
Patterns of MC development during early childhood (EC). Red lines indicate undesirable patterns whereas gray lines indicate other patterns.

**Table 1 ijerph-21-00190-t001:** Classification of MC patterns.

	Pattern	T0	T1	T2
**Stable MQ category over time**				
a.	**Continuously low ***	**Low**	**Low**	**Low**
b.	Continuously normal	Normal	Normal	Normal
c.	Continuously high	High	High	High
**Continuous change of direction in MQ category over time**				
d.	**Continuous decrease ***	**High**	**Normal**	**Low**
e.	Continuous increase	Low	Normal	High
**Fluctuating direction of change in MQ category over time**				
f.	**Decrease before stabilization *** Decrease before stabilization	**Normal** High	**Low** Normal	**Low** Normal
g.	**Decrease after increase *** Decrease after increase	**Low** Normal	**Normal** High	**Low** Normal
h.	**Decrease after stabilization *** Decrease after stabilization	**Normal** High	**Normal** High	**Low** Normal
i.	Increase before stabilization	Low Normal	Normal High	Normal High
j.	Increase after decrease	Normal High	Low Low or Normal	Normal or High Normal or High
k.	Increase after stabilization	Low Normal	Low Normal	Normal or High High

* Patterns classified as undesirable in **bold**.

**Table 2 ijerph-21-00190-t002:** Time-dependent characteristics of the study population (*n* = 1131, 50.2% boys) per timepoint (T0–T1–T2).

Characteristics	Timepoint		
	T0	T1	T2
Calendar year	2020	2021	2022
School grade ^1^	1 and 2	2 and 3	3 and 4
Age (years)			
Mean	5.35	6.36	7.26
SD	0.69	0.70	0.71
BMI (kg/m^2^)			
Mean SD	15.67 2.22	15.90 1.82	15.79 1.89
Childhood phase (%)			
Early childhood	81	33	3
Middle childhood	19	67	97

^1^: According to the Dutch school system.

## Data Availability

The data are available on request from the corresponding author at the Data Station Life Sciences of the Data Archiving and Networked Services (Dans). The data are not publicly available due to ethical reasons.

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
