# Peer review of "Changes in Motor Competence of 4–8-Year-Old Children: A Longitudinal Study"

_ijerph, 2024, doi:10.3390/ijerph21020190_

Round 1

Reviewer 1 Report

Comments and Suggestions for Authors

Dear authors,

Congratulations for the work done, I have very little suggestions.

Abstract

The abbreviation AST can be deleted, it is not used in the abstract.

To change motor competence for MC in the conclusion.

Keywords:

Do not repeat words in the title.

It will be better to present the words in alphabetical order.

Introduction

Figure captions should be placed below the figures.

To add the meaning of all abbreviations in the introduction: MQ. FMS, T1 …  tables and figures. And do not repeat the meaning: BMI….. Besides, some abbreviations: AMC, SES, KTK can be deleted because they are never used in the manuscript.

Table 2 can be smaller, on the other hand, figures 1 and 2 can be presented better.

Best Regards

Author Response

We thank you for your time and effort in reviewing our manuscript. The feedback has been very valuable in improving the content of the manuscript. My co-authors and I are pleased to submit our revised manuscript titled “Changes in Motor Competence of 4–8-year-old children: a longitudinal study." for reconsideration for publication. 

The changes are highlighted in the attached manuscript by using bold text, and our point-by-point responses are given in italics

Reviewer 1: 

Congratulations for the work done, I have very little suggestions.

Response: Thank you!

Comments (1) regarding the abstract: 

The abbreviation AST can be deleted, it is not used in the abstract. To change motor competence for MC in the conclusion.

Response 1: The AST abbreviation is removed and motor competence is being changed as suggested. Abstract (lines 21 and 27).

Comments (2) regarding the keywords:

Do not repeat words in the title. It will be better to present the words in alphabetical order.

Response 2: 

Comments (3) regarding the introduction:

Figure captions should be placed below the figures. To add the meaning of all abbreviations in the introduction: MQ. FMS, T1 …  tables and figures. And do not repeat the meaning: BMI….. Besides, some abbreviations: AMC, SES, KTK can be deleted because they are never used in the manuscript. Table 2 can be smaller, on the other hand, figures 1 and 2 can be presented better.

Response 3: Thank you for checking the consistent use of abbreviations and the technical specifications/presentation of the illustrations. Throughout our manuscript abbreviations have been removed. Illustrations have been updated/specified according to your suggestions (see figures and tables).  

Reviewer 2 Report

Comments and Suggestions for Authors

Dear Authors 

I have read your submitted article with interest, and underneath please find  suggestions and comments that I hope will help you with more convincing rationale for your study and better presentation. 

Title and abstract are informative and clearly presented. 

Introduction - this is the section that, in my opinion is a bit to generic and too short, and as such it requires more work from the Authors. 

Firstly, I think you should link your research to biological development and maturation processes in children (with emphasis on natural motor development trajectories in children of this age) and refer to times you held your research. Let me remind you that the time you carried out your research it was the time of COVID-19 pandemic and there is no reference to the lockdowns and other PA restriction. Also, this is important from educational perspective as children had to stay home, so the chances of stimulating the growth of motor competences was limited. 

Also, in the Introduction part, I think you could refer to similar longitudinal studies on the growth of motor competences some decades ago - to portray some historical background and clash it with present situation (increase use of modern technology and sedentary behaviours even in the youngest children age-category). 

Methods and research instruments are clearly defined and described with reliability coefficients. There are no ethical constrains and all consents have been acquired as it is requested in such works. 

Results are concise and well-presented, figures and tables are neat and readable. 

Discussion again, in my opinion to generic, just touching the surface of the problem and needs more in-depth analyses and linking it with general health and social situation world wide, not just in the Netherlands. It will bring more interest from the global audience. 

In Conclusions, you refer to the interventions needed in the area of MC of children, but this is not the finding of your study. I think, this is something that is missing in both Introduction and Discussion. Perhaps you could bring some information on effective interventions already existing in the topic research area. This would give a reader an idea of what has already been done in this area and what are the examples of best practice. And this will strengthen the rationale for your study. A reader needs to understand why did you pick up this topic for your research, what was your concern? 

References are up-to-date, but in my opinion is needs to be expanded. 

Author Response

We thank you for your time and effort in reviewing our manuscript. The feedback has been very valuable in improving the content of the manuscript. My co-authors and I are pleased to submit our revised manuscript titled “Changes in Motor Competence of 4–8-year-old children: a longitudinal study." for reconsideration for publication. 

The changes are highlighted in the attached manuscript by using bold text, and our point-by-point responses are given in italics

Reviewer 2

Title and abstract are informative and clearly presented. Thank you!

Introduction - this is the section that, in my opinion is a bit to generic and too short, and as such it requires more work from the Authors. 

Firstly, I think you should link your research to biological development and maturation processes in children (with emphasis on natural motor development trajectories in children of this age) and refer to times you held your research. Let me remind you that the time you carried out your research it was the time of COVID-19 pandemic and there is no reference to the lockdowns and other PA restriction. Also, this is important from educational perspective as children had to stay home, so the chances of stimulating the growth of motor competences was limited. 

Response 1: Thank you for the suggestions. We incoorporated a section regarding the average motor- and biological development of our target population (lines 42-56).  We also put more emphasis on the possible consequences of the COVID-19 pandemic in our manuscript. This is being done in our data analysis (lines 183-186), and in the discussion section (lines 277-294).   

Also, in the Introduction part, I think you could refer to similar longitudinal studies on the growth of motor competences some decades ago - to portray some historical background and clash it with present situation (increase use of modern technology and sedentary behaviours even in the youngest children age-category). 

Response 2: Again, thank you for the suggestion to improve our manuscript. In lines 72-76 a historical perspective is pointed out to give more information regarding the trends in MC development. It should be mentioned that high quality data are rare.

Methods and research instruments are clearly defined and described with reliability coefficients. There are no ethical constrains and all consents have been acquired as it is requested in such works.

Response 3: Thank you! However, we did make some adjustments for clarification (lines 175-189).

Results are concise and well-presented, figures and tables are neat and readable. Thank you!

Discussion again, in my opinion to generic, just touching the surface of the problem and needs more in-depth analyses and linking it with general health and social situation world wide, not just in the Netherlands. It will bring more interest from the global audience. 

Response 4: Besides the COVID-19 pandemic we have tried to create a more in-depth analyses to connect our findings with other factors like screentime and environmental influences (lines 264-276), and also socio-economic / cultural differences (339-343).

In Conclusions, you refer to the interventions needed in the area of MC of children, but this is not the finding of your study. I think, this is something that is missing in both Introduction and Discussion. Perhaps you could bring some information on effective interventions already existing in the topic research area. This would give a reader an idea of what has already been done in this area and what are the examples of best practice. And this will strengthen the rationale for your study. A reader needs to understand why did you pick up this topic for your research, what was your concern? 

Response 5: Thank you. In lines 329-336 we created insight in the benefits of motor interventions. How these can be beneficial for young children. 

References are up-to-date, but in my opinion is needs to be expanded. 

Response 6: References are revised and being expanded 

Reviewer 3 Report

Comments and Suggestions for Authors

Dear authors, first of all I would like to thank you for the opportunity to review this manuscript. I would like to thank you for the effort in carrying it out (longitudinal study), but I consider that the study does not have sufficient quality and interest to be published in IJERPH.

I would like to make some observations that could improve your work, always with the author's decision.

The objective of the present study was to describe and 14 quantify patterns of MC development from early childhood to middle childhood and to identify undesirable patterns.

The biggest problem with this study is that factors such as the habitual sports practice of schoolchildren throughout the process (start or dropout) have not been taken into account. This aspect may have considerably influenced the results obtained.

Nor have gender been taken into account in these differences or the body composition of the schoolchildren. These limitations can generate very important biases in the results obtained.

Author Response

We thank you for your time and effort in reviewing our manuscript. The feedback has been very valuable in improving the content of the manuscript. My co-authors and I are pleased to submit our revised manuscript titled “Changes in Motor Competence of 4–8-year-old children: a longitudinal study." for reconsideration for publication. 

The changes are highlighted in the attached manuscript by using bold text, and our point-by-point responses are given in italics

Reviewer 3

Dear authors, first of all I would like to thank you for the opportunity to review this manuscript. I would like to thank you for the effort in carrying it out (longitudinal study), but I consider that the study does not have sufficient quality and interest to be published in IJERPH.

I would like to make some observations that could improve your work, always with the author's decision.

The objective of the present study was to describe and 14 quantify patterns of MC development from early childhood to middle childhood and to identify undesirable patterns.

The biggest problem with this study is that factors such as the habitual sports practice of schoolchildren throughout the process (start or dropout) have not been taken into account. This aspect may have considerably influenced the results obtained.

Response 1: Thank you for your comments. Our goal was to describe the natural development of motor competence during the transition from early- to middle childhood. We therefore have chosen not to examine variables like type and amount of sports participation / physical activity / etc. Nor we included other determinants (e.g., percieved MC, enjoyment). Solely, the amount of PE lessons is explicitly mentioned since this is policy and thus uniform for our study population (lines 272-275).

Nor have gender been taken into account in these differences or the body composition of the schoolchildren. These limitations can generate very important biases in the results obtained.

Response 2: Thank you. These suggestions will help us for future analyses. We did examined the MQ scores for the drop-outs in our study and concluded that their MQ scores were not different compared with the participants with 3 measurements. 

Reviewer 4 Report

Comments and Suggestions for Authors

---General comment

The study does not present major innovations. In reality, it is in line with previous reports, as well pointed out by the authors. Still, literature on the topic is scarce, and follow-up studies are challenging. These factors must be considered positively.

---Specific comments

--Introduction

-The introduction is fluid, but the excess of acronyms clutters up some sentences. Check if they are all really necessary.

-I didn't find the hypothesis after the objective.

--Methods

-Line 85-87 - In this age group, motor development can drastically differ. Therefore, the authors need to explain in the article the reason for the age range, and, above all, whether this is a limiting factor in the study.

 -Figure 1 - The legend must appear below the figure, providing the meaning of its acronyms (T0, T1, etc.). Additionally, I recommend colors in the image to bring the figure to life.

 -Line 124 - Here and throughout the manuscript change weight to body mass

Author Response

We thank you for your time and effort in reviewing our manuscript. The feedback has been very valuable in improving the content of the manuscript. My co-authors and I are pleased to submit our revised manuscript titled “Changes in Motor Competence of 4–8-year-old children: a longitudinal study." for reconsideration for publication. 

The changes are highlighted in the attached manuscript by using bold text, and our point-by-point responses are given in italics

Reviewer 4: 

General comment

The study does not present major innovations. In reality, it is in line with previous reports, as well pointed out by the authors. Still, literature on the topic is scarce, and follow-up studies are challenging. These factors must be considered positively. Thank you!

---Specific comments

--Introduction

-The introduction is fluid, but the excess of acronyms clutters up some sentences. Check if they are all really necessary. 

Response 1: We have revised the manuscript for abbreviations and acronyms, maintining and/or improving the quality of sentences.

-I didn't find the hypothesis after the objective.

Response 2: Thank you for this remark. We incorporated our hypothesis in lines 102-117. 

--Methods

-Line 85-87 - In this age group, motor development can drastically differ. Therefore, the authors need to explain in the article the reason for the age range, and, above all, whether this is a limiting factor in the study.

Response 3: Thank you for this consideration. To put our results in perspective we have pointed out the 'normal' stages of motor- and biological development (lines 42-56) also from the perspective of the proficiency barrier. We also pointed out the importance of MC development from early childhood since this can help professionals and policy makers to intervene at the transition from early- to middle childhood (lines 327-336). 

 -Figure 1 - The legend must appear below the figure, providing the meaning of its acronyms (T0, T1, etc.). Additionally, I recommend colors in the image to bring the figure to life.

Response 4: Thanks again for the suggestions. All tables and figures have been revised (colors have been added). 

 -Line 124 - Here and throughout the manuscript change weight to body mass

Response 5: This has been done!

Round 2

Reviewer 2 Report

Comments and Suggestions for Authors

I am glad that you have incorporated most of the reviewers' suggestions and this text is much clear and reads better now. 

Reviewer 3 Report

Comments and Suggestions for Authors The authors have made enough changes for the article to be accepted in its current form.

Reviewer 4 Report

Comments and Suggestions for Authors

I have no further comments.